SOFTWARE

# MINT: A toolbox for the analysis of multivariate neural information coding and transmission

**Gabriel Matías Lorenz**[1,2,3☯], **Nicola Marie Engel**[1☯], **Marco Celotto**[1,4], **Loren Koçillari**[1,5], **Sebastiano Curreli**[2], **Tommaso Fellin**[2], **Stefano Panzeri**[1]*

**1** Institute for Neural Information Processing, Center for Molecular Neurobiology (ZMNH), University Medical Center Hamburg-Eppendorf (UKE), Hamburg, Germany, **2** Optical Approaches to Brain Function Laboratory, Istituto Italiano di Tecnologia, Genova, Italy, **3** Department of Pharmacy and Biotechnology, University of Bologna, Bologna, Italy, **4** Department of Brain and Cognitive Sciences, Picower Institute for Learning and Memory, Massachusetts Institute of Technology, Cambridge, Massachusetts, United States of America, **5** Department of Neurophysiology and Pathophysiology, University Medical Center Hamburg-Eppendorf (UKE), Hamburg, Germany

☯ These authors contributed equally

* s.panzeri@uke.de

## Abstract

Information theory has deeply influenced the conceptualization of brain information processing and is a mainstream framework for analyzing how neural networks in the brain process information to generate behavior. Information theory tools have been initially conceived and used to study how information about sensory variables is encoded by the activity of small neural populations. However, recent multivariate information theoretic advances have enabled addressing how information is exchanged across areas and used to inform behavior. Moreover, its integration with dimensionality-reduction techniques has enabled addressing information encoding and communication by the activity of large neural populations or many brain areas, as recorded by multichannel activity measurements in functional imaging and electrophysiology. Here, we provide a Multivariate Information in Neuroscience Toolbox (MINT) that combines these new methods with statistical tools for robust estimation from limited-size empirical datasets. We demonstrate the capabilities of MINT by applying it to both simulated and real neural data recorded with electrophysiology or calcium imaging, but all MINT functions are equally applicable to other brain-activity measurement modalities. We highlight the synergistic opportunities that combining its methods afford for reverse engineering of specific information processing and flow between neural populations or areas, and for discovering how information processing functions emerge from interactions between neurons or areas. MINT works on Linux, Windows and macOS operating systems, is written in MATLAB (requires MATLAB version 2018b or newer) and depends on 4 native MATLAB toolboxes. The calculation of one possible way to compute information redundancy requires the installation and compilation of C files (made available by us also as pre-compiled files). MINT

**Data availability statement:** All data and code used to produce this paper is made publicly available. All code is downloadable in source code (https://github.com/panzerilab/MINT with DOI doi.org/10.5281/zenodo.13998526) and is licensed under GNU GPLv3. The neural CA1 data are attached to this submission as Supplemental Material.

**Funding:** This work was supported by the Simons Foundation for Autism Research Initiative (SFARI) grant number 982347 (to SP) and the NextGenerationEU FAIR grant number PE0000013 (to TF). The Funders had no role in study design, data collection and analysis, decision to publish, or preparation of the manuscript.

**Competing interests:** I have read the journal's policy and the authors of this manuscript have the following competing interests: Stefano Panzeri is a member of the editorial board of PLoS Computational Biology. No other conflict of interest is declared by any other author.

is freely available at https://github.com/panzerilab/MINT with DOI doi.org/10.5281/zenodo.13998526 and operates under a GNU GPLv3 license.

## Introduction

Brain functions are based on the ability of groups of neurons or brain areas to encode, process and transmit information [1,2]. Consequently, information theory [3], the mathematical theory of communication, has deeply influenced the conceptualization of brain operations. It has become a method of choice to analyze neural activity because of its many advantages [4–8]. It provides single-trial measures of how neural activity encodes variables important for cognitive functions such as sensory stimuli, and it is thus more relevant for single-trial behavior than trial-averaged measures. It captures contributions of both linear and non-linear interactions between variables at all orders, and thus allows hypotheses-free measures of information encoding that place upper bounds to the performance of any decoder. Because of its generality, it can be applied to any type of brain activity recordings. Also, it facilitates direct comparisons between the predictions of normative neural theories and real neural data [6,9].

Earlier work using information theory to analyze empirical neural data has focused on low-dimensional measures of neural activity such as single neurons, small neural populations or aggregate measures (LFPs, M/EEG, fMRI). These studies have considered only how information is encoded in neural activity, regardless of how it may be used downstream. Such seminal studies have demonstrated, e.g., how the temporal structure of neural activity (from single-neuron spike timing to network oscillations [10–16]) contributes to sensory encoding, or how neural mechanisms such as adaptation contribute to brain information processing [8,17].

Over the last decade, neuroscience has seen major progress in the ability to record simultaneously the activity of many neurons and/or brain areas. These advances have driven the development of novel information theoretic analytical tools to investigate how information processing emerges from the interaction and communication among neurons or areas. Studies have provided multivariate information tools to individuate when synergy and redundancy arise in small populations, or to understand the mechanisms for generating redundancy and synergy, for example to characterize how correlations between the activity of different neurons shape information processing [1,18–22]. Recent work has also coupled information theory with dimensionality-reduction techniques to study how information is encoded in populations of tens to hundreds of cells [23–30]. Other studies have developed multivariate information theory to quantify transmission, rather than encoding, of information across neurons or areas [31–41]. These methods measure the overall or stimulus-specific information exchanged between simultaneously recorded neurons and areas and determine whether transmission relies on synergistic integration of information across nodes. Another major direction of progress has been in recording neural activity during behavior [42]. To support the growing interest on how neural computations shape behavior, information theory has produced tools to characterize

the multivariate simultaneous relationship between sensory stimuli, neural activity and behavioral output to enable quantifying the impact on behavior of the information encoded in a certain area or population [24,43,44].

While the use and dissemination of information theoretic algorithms has been aided by software toolboxes [45–61], no toolbox yet provides a comprehensive implementation of tools to compute both information encoding and transmission, to break down information into components reflecting the effect of interactions and to quantify behavioral or downstream relevance of the encoded information (see Table A in S1 Appendix). To fill these gaps and address the need to collect organically these tools in a format that allows immediately multiple analyses, here we introduce a new Multivariate Information in Neuroscience Toolbox (MINT). MINT provides a comprehensive set of information theoretic functions (including Shannon Entropy and Mutual Information, directed information transmission measures, information decompositions) and estimators (binned probability estimators, limited-sampling bias corrections). The implemented information-theoretic functions are detailed in S1 Appendix. What they compute, and how they can be used in neuroscience is summarized in Table 1. The accuracy and applicability of these algorithms has been validated and demonstrated extensively with both discrete neural variables, such as spikes in electrophysiological recordings [8,13,14,62–64], and continuous neural variables, e.g., LFP, M/EEG, fMRI and calcium traces [24,30,65–67].

Importantly, as we demonstrate with examples, combining these multivariate tools enables addressing questions that cannot be addressed with a single tool. For example, combining tools to identify the specific contribution of correlations to population encoding or the amount of encoded information that informs behavior with dimensionality-reduction techniques allows understanding how large neural populations influence behaviors. Combining information encoding tools with content-specific information transmission tools can reverse engineer information flow in neural networks with unprecedented understanding. We thus anticipate that MINT will lead to uncover numerous new insights into neural information processing.

## Design and implementation

MINT is written in MATLAB (version 2018b or newer) and depends on the Statistics and Machine Learning, Optimization, Parallel Computing and Signal Processing Toolboxes. MINT takes as input neural data (array of neural activity recorded in each trial) and task variables (sensory stimuli or behavioral responses presented or produced in each trial). It outputs information values and their null-hypothesis values for computing statistical significance. Fig A in S1 Appendix illustrates MINT functions, options, and core routines.

MINT computes Entropy (H.m), which measures neural variability; and Mutual Information (MI.m), which measures information encoding (Fig 1B and 1C). It computes the Information Breakdown of Shannon Information into contributions due to correlations between neurons [18,20,21,68,69]. It also computes Partial Information Decompositions (PID) [70, 71] of the information about a target variable carried by two or more source variables into unique, synergistic and redundant information (Fig 1C). Computation of PID requires specifying a redundancy measure, which can be selected by the user among options [52,70,72,73] with complementary advantages. (Redundancy of [52] requires either a MATLAB-compatible C compiler or pre-compiled files made available by us for Windows 11, macOS, and Linux Debian). MINT computes additional functions of neuroscientific value: Intersection Information (II, function II.m; Fig 1B, see [44]), the amount of stimulus information in neural activity that is used to inform behavior; Transfer Entropy (TE, see [74]) and Feature-Specific Information Transfer (FIT, see [36]), which measure overall and stimulus-feature-specific information transmission between nodes of neural networks (TE.m, FIT.m and cFIT.m; Fig 1D).

The information quantities depend on the probabilities of task variables (e.g., presented sensory stimuli) and neural responses. MINT implements the *direct method* [14,75] estimator based on discretizing neural responses and task variable values and computing the empirical occurrences across experimental trials of the discrete or binned responses. These estimators have been widely used in neuroscience information theoretic studies, because neural spiking activity is intrinsically discrete and is usually quantified as the number of spikes emitted in one- or multiple-time windows of

**Table 1.** *Glossary of main information theoretic functions.*

| Function | What is Computed | What it is Used for |
|---|---|---|
| **Entropy (H)** | Variability of random variable X | Assess variability of neural activity |
| **Mutual Information (MI)** | How well an ideal observer can predict X from single-trial observations of Y | How well activity of population of neurons encodes info about task variables (e.g., stimuli) |
| **Information Breakdown** | MI about X carried by a multivariate Y, broken down into contributions arising from correlations between different dimensions of Y | How correlations between multivariate neural activity (e.g., activity of populations of neurons) contribute to population-level encoding of task variables (e.g., sensory stimuli) |
| **Partial Information Decomposition (PID)** | Decomposes MI carried by multidimensional Y about X into unique info about X carried by each element of Y, synergistic info found only in the interactions among elements of Y, and redundant info shared among elements of Y | Whether groups of neurons or brain areas carry synergistic or redundant info about task variables (e.g., stimuli) or about activity of other neurons or brain areas |
| **Redundancy-Synergy Index (RSI)** | Quantifies whether the effect on encoding info about X of interactions between different variables within a multi-dimensional Y is predominantly redundant or synergistic | Whether groups of neurons or brain areas carry predominantly synergistic or redundant info |
| **Intersection Information (II)** | Info about X carried by Y which is used to inform Z about X | How much of the info about a task variable (e.g., sensory stimulus) encoded in neural activity is used to inform behavioral reports (e.g., choices) |
| **Transfer Entropy (TE)** | MI about past activity of sender X found in future activity of receiver Y, conditioned on past activity of Y | Measures overall transmission of info between nodes of a neural network |
| **Feature Specific Information Transfer (FIT)** | MI about feature S presently encoded by Y redundant with MI about S previously encoded by X and unique w.r.t. MI about S previously encoded by Y | Measures info transmission between nodes of a neural network of info about a specific feature of task variables (e.g., sensory stimuli) |
| **Conditional TE (cTE) and Conditional FIT (cFIT)** | Versions of TE and FIT with info flow conditioned or unique w.r.t. another variable | Measure overall or feature specific transmission of info between network nodes discounting info possibly passing through other nodes |
| **Supervised dimensionality reduction (decoding)** | Project data onto lower-dimensional space using labeled data to optimize decoding | Intermediate step for information calculations with large neural populations |
| **Unsupervised dimensionality reduction** | Project data onto lower-dimensional space using unlabeled data to optimize data explainability | Intermediate step for info calculations with large neural populations |
| **Limited-sampling bias correction** | Produces unbiased info estimates unaffected by the limited-sampling bias | Needed to obtain more accurate info estimates in all practical situations |
| **Hierarchical Data shuffling** | Random permutations of data used to create null distribution for statistical testing for assessing the role of certain data features in info encoding | Needed for statistical testing in all applications. Useful to assess role of spike timing or correlation between neurons by comparing info values obtained with these features preserved or shuffled |

This table reports a short explanation of what the implemented information theoretic quantities compute and for what type of applications they may be used. X, Y, Z denote random variables.

interest [14,75]. The direct method captures the information carried by spike counts very precisely (Fig C in S1 Appendix). Because they are simple and do not make assumptions about the probability distributions, discretized estimators have been used to compute information also from continuous-valued aggregate measures of neural activity such as LFP, M/EEG, fMRI [11,65,66,76] or continuous-valued behavioral variables [77]. If the scientific question at hand needs PID in addition to Shannon information and the data are not Gaussian, then discrete or discretized approaches are advised (as non-discrete non-parametric estimators are available only for Shannon information and entropy). MINT provides binning functions to discretize analogue data (equi-spaced or equi-populated binning, binning with user-defined bin edges, and possibly automated determination of bin numbers [78,79]).

Any real experiment only yields a finite number of trials from which probabilities must be estimated. Finite sampling when using direct methods leads to a systematic error (bias) in information estimates (Fig 1E), which can be as big as the

true information values. Thus, plugin estimators (based on just plugging in probabilities into the information equations) are biased, and bias corrections methods need to be added to it and are essential for practical neuroscience applications. Six such well-established methods are included in MINT [80–85]. These methods, along with binning, parallelization options and other features are user-specified in an input structure (opts). Information (function MI.m) is computed by default with the plugin method, as it preserves all information available in the discretized neural activity. We recommend its use for small-dimensional (up to N = 2 or 3) neural response (e.g., responses of populations of up to 2–3 neurons) as its estimates from datasets of realistic sizes can be still effectively corrected for the limited-sampling bias (Fig C in S1 Appendix).

Alternatively, probability estimators suited for real-valued data [86–88], such as nearest-neighbors or kernel methods, can be used to estimate information and are available in MINT by specification in the input structure (opts). These methods also work well for low-dimensional data.

Neither these estimators nor the direct method, however, work on their own when considering high-dimensional neuronal responses (such as the activity of populations of many neurons), as the curse of dimensionality prevents the direct sampling of the joint response probabilities from high dimensional data (Fig C in S1 Appendix). We thus provide additional pipelines, recommended for high-dimensional neural responses such as the activity of large neural populations, that compute information from the empirical neural response probabilities but after reducing the dimensionality of neural population activity [24]. These *dimensionality-reduction pipelines* include supervised methods (Support Vector Machines, SVMs and Generalized Linear Models, GLMs) which reduce the dimensionality by providing decoding or posterior probabilities of the task variables given the single-trial neural population activity (Fig 1A) and allow reliable estimations with small datasets (Fig C in S1 Appendix). We also provide unsupervised methods (Non-negative Matrix Factorization, NMF [89]; Principal Component Analysis, PCA) which reduce dimensionality individuating small numbers of dimensions with the highest explanation power of neural activity. Supervised dimensionality-reduction algorithms that individuate the directions in neural activity space with most discriminability of the task variable (e.g., SVM) may be in general better suited than unsupervised algorithms individuating dimensions that target best reconstruction of the spike trains (e.g., PCA, NMF) when the most information is not encoded in the direction with most variations in neural activity space (Fig C in S1 Appendix).

MINT provides all these dimensionality-reduction techniques with native MATLAB functions, but it also allows easy interfacing with external libraries (e.g., libsvm [90] and glmnet [91]) (Fig B in S1 Appendix). Importantly, these dimensionality-reduction tools can be coupled (Fig 1F) with MINT's *Hierarchical Shuffling* tool (hShuffle.m) which can disrupt, by trial shuffling, specific features of population activity (such as response timing or correlations between neurons) to probe their contribution to information processing [24,92].

When deciding which estimator to apply to a given dataset, we recommend users to test different algorithms on synthetic data that match essential features of the experiments (e.g., discrete spike counts or continuous signals, number of trials and data dimensionality, information levels) and choose what suits best. MINT provides a simulator of correlated or uncorrelated neural population spike train activity that can be used for this purpose.

## Results

We illustrate how to use MINT to address highly topical neuroscientific questions, emphasizing the utility of using synergistically multiple algorithms, allowed by MINT. In all examples, we use the limited-sampling bias corrections and hierarchical data shuffles of MINT, as they are essential for empirical data analyses.

### Computing the role of interactions between neurons in information encoding

An important question in neuroscience is whether and how the functional interactions (measured as activity correlations) between neurons enhance or limit information encoding in neural populations [1,93]. Several information theoretic methods have been developed to address complementary aspects of this question [18–22,68,70,92,94]. Here we illustrate what we gain from their combined use enabled by MINT.

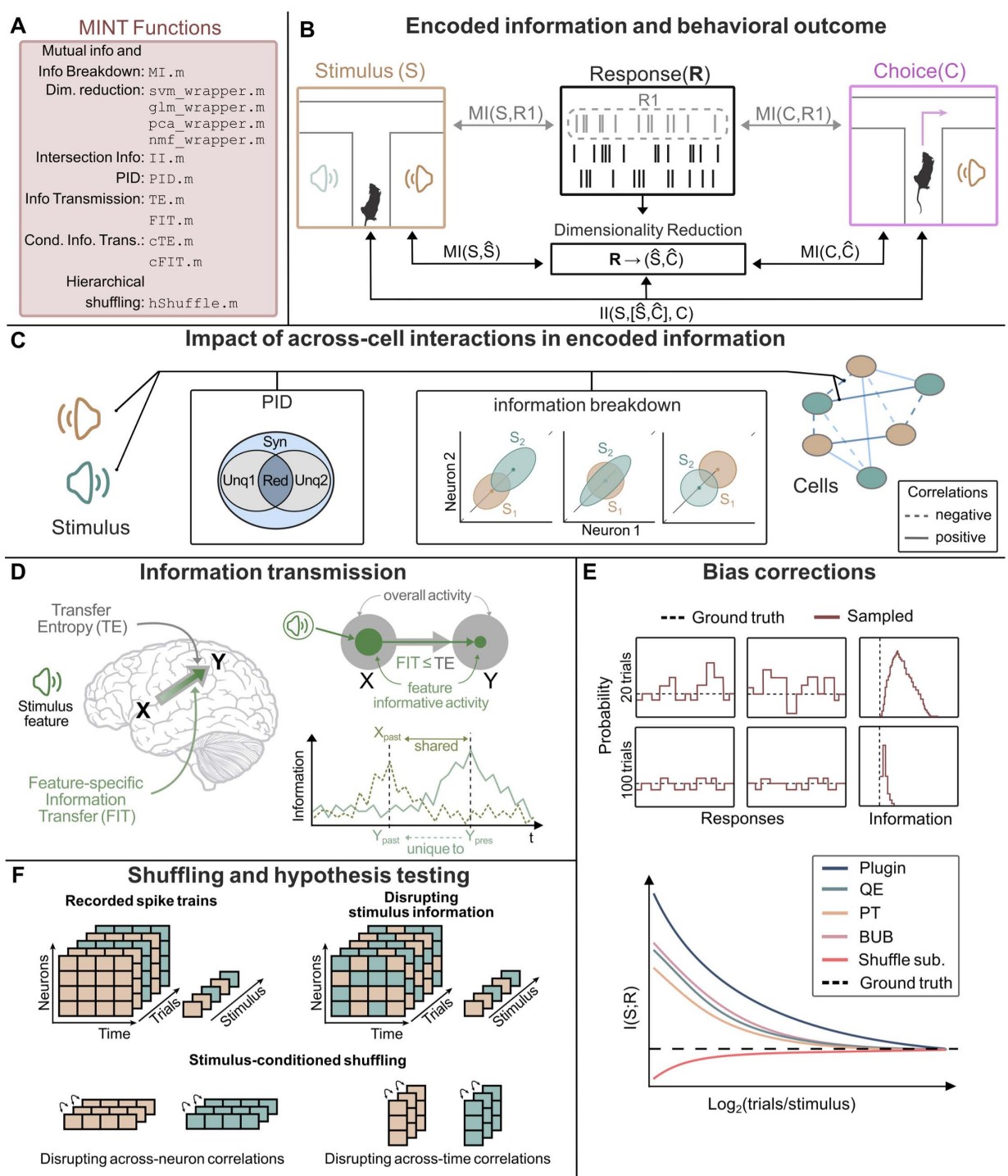

**Fig 1. Overview of MINT.** A: List of main MINT functions. B: MINT provides multivariate information theoretic functions to quantify the amount of information that single neurons or neural populations carry about task-relevant variables (e.g., sensory stimuli or behavioral choices). These methods are based on either direct-method calculation of information from the discretized probabilities (ideal for small neural populations but not scalable with population size), estimation through other techniques such as Kernel-based methods that can operate on real-valued data, or by using supervised or unsupervised dimensionality-reductions techniques to approximate high-dimensional neural population response probabilities with probabilities in lower-dimensional spaces (scalable with population size). It also provides tools to quantify how much of the information encoded by neural activity is used to inform behavior. C: MINT has multiple functions to compute, in small or large populations, how interactions between the activity of different neurons

shape information encoding and create synergy or redundancy. D: MINT has tools to compute transfer of information from one neural population or brain region to another. It can compute both the total or stimulus-specific information transmitted between two nodes, with the option of conditioning over the activity of other nodes. E: MINT has tools to correct plugin estimators for the limited-sampling bias, an essential tool for analysis of empirical neuroscience data. F: MINT has a set of hierarchical permutation algorithms that provide null hypothesis testing for significance of information encoding and information transmission and for the impact of correlations across neurons or time. Mouse sketch is modified from doi.org/10.5281/zenodo.3925917, brain sketch is modified from doi.org/10.5281/zenodo.3925989, and speaker sketch is modified from svgrepo.com/svg/506329/speaker-2. All resources are licensed under CC BY 4.0 (creativecommons.org/licenses/by/4.0).

We consider how a population of N neurons encodes information about a stimulus variable S. For neuron pairs (N = 2), we computed the population information (Mutual Information between stimulus and the joint neural population response) with the direct method that estimates information directly form the empirical discretized response probabilities (see Design and implementation). The overall effect of interactions between neurons is expressed by the Redundancy-Synergy Index (RSI), the difference between the population information and the sum of single-neuron stimulus information [19,95]. Positive (negative) RSI indicates predominantly synergistic (redundant) interactions. Contributions of synergy and redundancy can be separated using PID [71,96]. The Information Breakdown [20–22] shows how RSI arises from interactions between neurons, by breaking down RSI into components $I_{sig-sim}$ (contribution of the similarity across neurons of trial-averaged responses to different stimuli, see also [19]), $I_{cor-ind}$ (contribution of the interplay between the signs of signal similarity and of noise correlations, defined as correlations between neurons in trials to the same stimulus), and of $I_{cor-dep}$ (quantifying information added by the stimulus-modulation of noise correlations, or, equivalently, bounding the information lost when using decoders trained without considering correlations [18,69]).

These small-population direct information calculations have the advantage of not making assumptions about decoding mechanisms, but do not scale up to large populations because of the curse of dimensionality [82]. Population information can be obtained by estimating probabilities in the reduced space of the stimuli decoded from single-trial neural activity. These estimates scale well with population size and can be computed robustly with small datasets (Fig C in S1 Appendix). However, specific decoders may severely underestimate total information in neural activity (see Fig C in S1 Appendix), especially when the decoder does not operate on the features of neural activity that carry most information. We illustrate below how MINT allows determining the role of correlations in population coding by comparing decoders that do or do not use information in correlated activity and by leaving intact or removing information in correlated activity using hierarchical shuffling tools [1,24,92].

We illustrate these methods first by simulating the activity of N = 20 neurons responding to two stimuli. In the first simulated scenario (Fig 2A), only correlations between activity of different neurons, but not the single-neuron activities, are stimulus-modulated and thus encode stimulus information. The single cell information is zero, but the pairwise population information is not. Positive RSI arises because of large synergy with negligible redundancy. The Information Breakdown reveals that all the synergistic information is due to stimulus-dependent correlations. Population decoding with SVM of the N = 20 neurons reveals that large-population information can be accessed exclusively with a non-linear decoder, and that shuffling correlations destroys all information, confirming it is exclusively encoded by correlations.

In the second simulated scenario (Fig 2B), information is encoded by single cells, correlations are only weakly stimulus-modulated, all neurons have equal stimulus tuning (responding more strongly to stimulus 2), and noise correlations are positive. In this configuration, redundancy is created (all neurons have the same trial-averaged response profiles to the stimuli) and correlations reduce information (they are elongated along the axis separating the mean firing rates of individual neurons and thus increase the overlap between the stimulus-specific distributions of neural activity) [20]. Negative RSI arises because of larger redundancy (created by so called signal similarity expressing the similarity of tuning to stimuli of individual neurons) than synergy (created by the small but present stimulus-modulation of correlations). Information Breakdown analysis reveals that indeed information is more redundant because the signal-noise similarity (captured by $I_{sig-sim}$ and $I_{cor-ind}$) is larger than the small stimulus-dependent correlations $I_{cor-dep}$. In the large (N = 20) population most

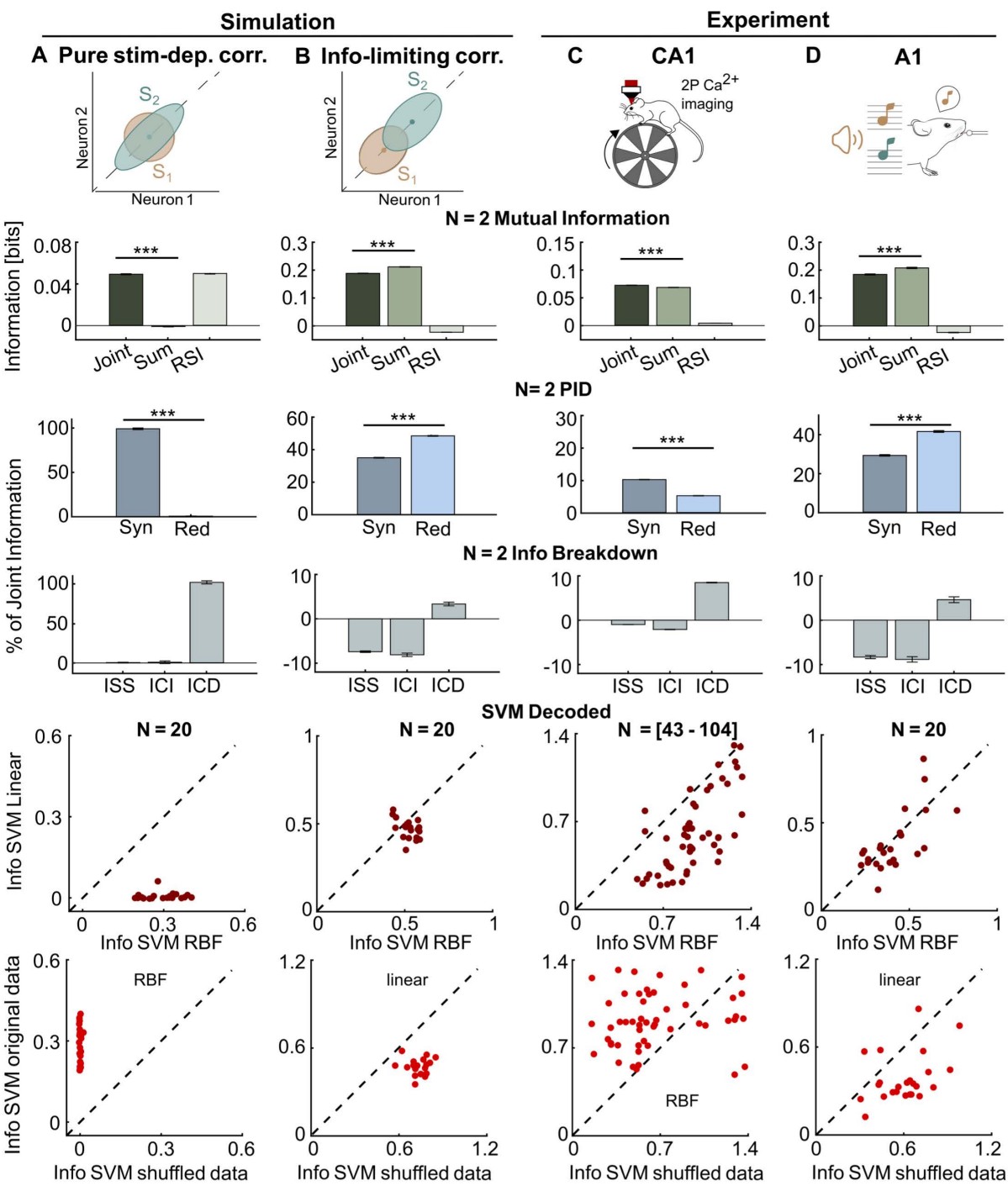

**Fig 2. Assessing the role of correlations among neurons in neural population encoding.** In each column we consider analysis of a different dataset. **A**: simulated population of N = 20 neurons which carry information only by stimulus-dependent correlations, with no stimulus information provided by single-neuron firing rate modulation. **B**: simulated population of N = 20 neurons which carry information by single neuron firing modulations and which have information-reducing correlations. C: CA1 recordings of N = [43–104] neurons over n = 11 sessions during spatial navigation of a linear track in virtual reality. D: A1 recordings of N = 20 neurons over n = 12 sessions during tone presentation. In each row, we plot from top to bottom: direct calculation of information for neuron pairs and of sum of single neuron information; direct calculation of redundancy-synergy index (RSI), of synergy and redundancy separately and of the Information Breakdown components for neuron pairs; calculation of encoded information of the whole population using the information in the confusion matrix of an SVM decoder (linear or RBF), computed either on the real population responses (which contain correlations

between neurons) or pseudo-population "shuffled" response obtained collecting randomly permuted trials to the same stimulus (shuffling removes correlations at fixed stimulus). In columns A-B we compute Shannon Information between neural activity and the identity of the two simulated stimuli. In column C-D we compute Shannon Information between neural activity and the identity of the presented tone (S = 2 different tones) or the spatial location of the mouse (binning locations into S = 12 equi-distant spatial bins), respectively. In column C, direct measures of pairwise information were obtained with R = 2 equi-populated bins (appropriate for this dataset consisting of non-deconvolved calcium fluorescent traces). In column D, direct measures of pairwise information were obtained with R = 3 bins, done by capping to 2 spike counts (appropriate for this dataset consisting of calcium signals deconvolved to estimate firing rates and activity counted in short windows). In each panel we plot mean ± SEM (for simulated data in panel A-B: over n = 190 neural pairs and n = 10 simulation repetitions for the direct information calculations; over n = 5 different data folds and n = 10 simulation repeats for the decoding information values; for CA1 data in Panel C: over n = 36158 simultaneously recorded neuron pairs for the direct information calculations, and over n = 11 recording sessions and n = 5 trial folds for the decoding information values; for A1 data in Panel D: over n = 2280 simultaneously recorded neuron pairs for the direct information calculations, and over n = 12 recording sessions and n = 2 trial folds for the decoding information values). Symbols *, **, *** denote two-tailed p < 0.05, p < 0.01, p < 0.001 respectively, computed with paired t-tests. See SM6.1, SM7.2 and SM7.3 in S1 Appendix for details of simulations and real data analysis. Mouse sketch in Panel D is modified from doi.org/10.5281/zenodo.3925985, speaker sketch is modified from svgrepo.com/svg/506329/speaker-2 and music note sketch is modified from svgrepo.com/svg/458908/sound. All resources are licensed under CC BY 4.0 (creativecommons.org/licenses/by/4.0).

information can be accessed with a linear SVM, with the non-linear SVM adding relatively little, and noise correlations reduce information (shuffling them away increases information).

We then applied the same analyses to two real neural datasets. We first analyze encoding of the mouse position (within a linear track) by populations of 43–104 simultaneously recorded neurons from the CA1 region of the mouse hippocampus [27] (Fig 2C). With the pairwise analysis, PID shows that both synergy and redundancy are present, but synergy is larger and the Information Breakdown shows that this is due to modulation of the noise correlations strength with the position ($I_{cor-dep}$ ~10% of the pairwise information). Using a nonlinear decoder of the whole population increases information by ~80% over what could be achieved with linear decoders, and shuffling data to destroy correlations decreases the nonlinearly decoded information by ~80%, revealing a large effect of hippocampal noise correlations in position encoding by large neural populations, whose size could not be inferred by neuron pairs analysis.

We then analyzed encoding of sound intensity by populations of 20 neurons simultaneously recorded from the mouse auditory cortex (A1) during pure-tone sound presentation (Fig 2D). These networks were selected, among all recorded neurons, based on their encoding of task-relevant information in [97]. With the pairwise analysis, PID shows that both synergy and redundancy are present, but redundancy is larger. Information Breakdown analysis shows that this is due to negative $I_{sig-sim}$ (neuron pairs have similar tuning to the stimuli) and $I_{cor-ind}$ (most neural pairs have also positive correlations), with $I_{cor-dep}$ contributing much less. Decoding whole-population activity with a nonlinear SVM did not increase the information decoded with a linear SVM (stimulus-dependent correlations were weak), and shuffling away noise correlations increases information substantially (thus correlations strongly reduced information).

Together, these results illustrate the power of combining MINT tools to understand deeply how interaction between neurons shape neural population coding.

## Computing how stimulus information encoded in neural population activity informs behavioral discriminations

Traditional approaches to neural information encoding of sensory stimuli have focused solely, as in the above examples, on how neurons or populations encode information about these stimuli. However, it could be that little or none of the information they encoded is actually utilized to inform behavior. It is thus important to have instruments to understand how much information in neural activity contributes to behavior.

Intersection Information (II) measures how much of the sensory information encoded in neural population activity is read out to inform behavior (Fig 3A; see sketch in Fig 1B), and is computed with PID (using the tri-variate probabilities of stimuli, neural activity and behavioral choices) as the component of neural information that is both about stimulus and choice [24,43,44]. To demonstrate its use, we applied it to analyze the activity of populations of neurons recorded with 2-photon calcium imaging in mice in auditory cortex (A1) during pure-tone perceptual discrimination [97] (Fig 3B).

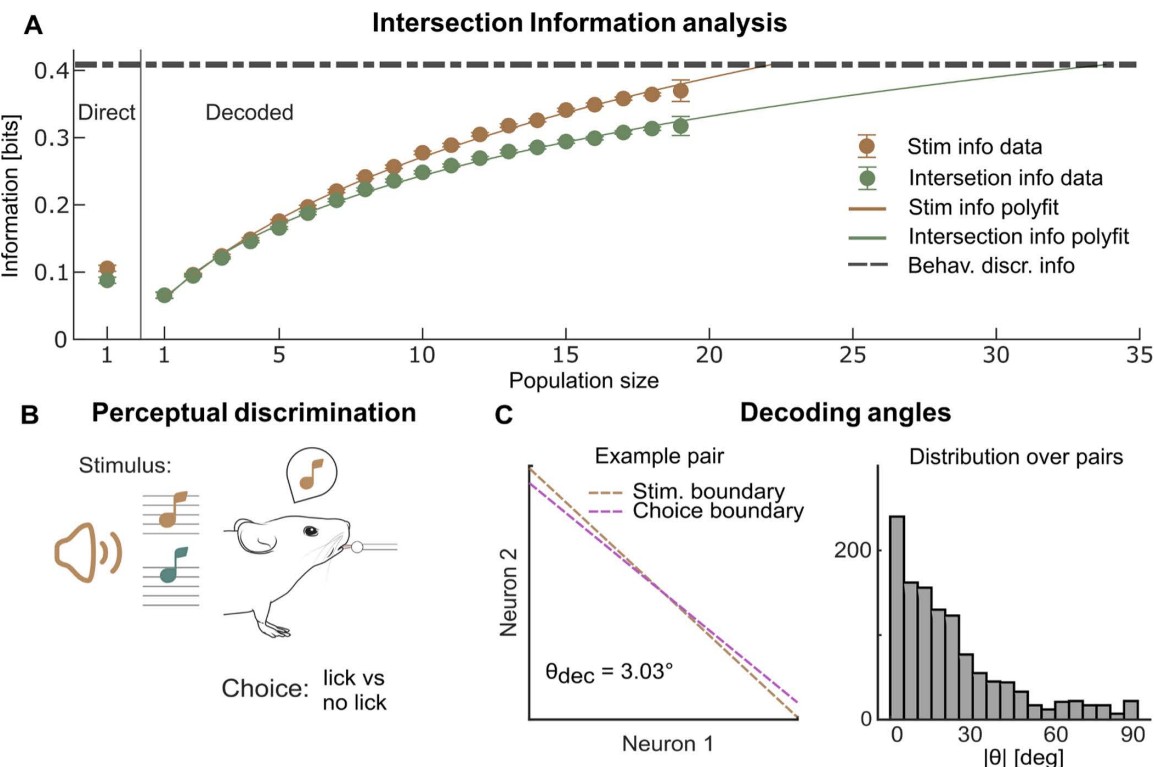

**Fig 3. Stimulus and Intersection information coding in populations of cortical auditory neurons during a tone discrimination task.** A: Stimulus information and Intersection Information encoded in neural activity recorded during a sound tone discrimination task. Left: single cell estimates using the direct method. Right: estimates of the information quantities using a RBF SVM (2-fold cross validation) as function of the population size. We plot the mean±SEM over all n = 12 Field of Views and over all folds and over all subpopulations used. For population sizes N = 2–18, for >100 independent sub-populations can be obtained, we shortened computation time using only n = 100 randomly sampled subpopulations. For population size N = 1 and 19, we used all the n = 20 different subpopulations available. For the direct information calculation, we used 3 bins for 0 spikes, 1 spike and any value above 1. For all information analyses, we used the shuffle subtraction to correct for the limited-sampling bias. The dashed horizontal line plots the averaged information needed to explain behavioral discrimination accuracy (computed as the information between stimulus and choice). Full lines show log-polynomial fits to the dependence of stimulus and intersection information on population size. The population size with information sufficient to explain behavioral discrimination accuracy is the x-axis intercept of the point at which the fit lines cross. B: Schematic of the behavioral task in mice used when recording the data analyzed in this figure. C: Stimulus and choice boundary computed with MINT in the space of paired neural activity for one example neural pair in the dataset. The value of the angle between the two axes is reported in the inset. Right: distribution of the absolute value of the angle between the stimulus and choice boundaries for the n = 2280 neural pairs in this dataset. See SM6.2 and SM7.3 in S1 Appendix for details of simulations and real data analysis. Mouse sketch is modified from doi.org/10.5281/zenodo.3925985, speaker sketch is modified from svgrepo.com/svg/506329/speaker-2 and music note sketch is modified from svgrepo.com/svg/458908/sound. All resources are licensed under CC BY 4.0 (creativecommons.org/licenses/by/4.0).

We first considered information encoded in single neurons, computed with the direct method. If the readout of the stimulus information in neural activity was optimal (respectively, completely suboptimal), II would equal the stimulus information, (respectively, be zero). We found that for single neurons, II was ~90% of the total single-neuron stimulus information, showing that information encoded by these neurons is not read out optimally but still efficiently.

For sampling reasons explained above, the direct calculation of II can be done for small (N = 1–3), but not for large populations. How can we use II to address how information relevant to behavior scales with population size? Specifically, how large must a population be to account for perceptual discrimination ability? To answer this, in MINT we combined II with dimensionality-reduction techniques. In this application, we used an SVM to compress neural activity (using svm_wrapper.m before II.m). This compression loses some information (the information values obtained with the plugin method are

~20% higher than the single cell values obtained with SVM decoders; Fig 3A). However, II population information computed with SVM decoders are scalable and data-robust (Fig C in S1 Appendix). Computing how information scales with population size (Fig 3A) shows that as population size increases, the gap between stimulus information and II widened. This means that behaviorally-relevant information is more redundant across neurons compared to information that is not used to inform behavior, confirming the usefulness of redundancy for behavioral readout [98]. Had we considered only stimulus information, we would have incorrectly concluded that ~23 such neurons are sufficient to account for the mouse discrimination performance (Fig 3A). However, taking intersection information into account reveals that ~34 such neurons are instead needed to fully account for the perceptual discrimination ability, as not all stimulus information encoded in neural populations is read out (Fig 3A).

We endowed MINT with instruments to characterize neural mechanisms of readout. Suboptimality may arise because of a misalignment between how information is encoded and how the brain reads it out to inform choices [43]. MINT returns the axes in neural activity space trained to discriminate between stimuli and the axes trained to discriminate different choices (using svm_wrapper.m, see Fig D in S1 Appendix for examples on simulated data). Computing decoding angles of pairs of A1 neurons (Fig 3C) shows that most pairs had a small but non-zero misalignment between stimulus and choice decoders, which explains the efficient but sub-optimal readout.

In sum, combining Intersection Information and dimensionality reduction can give precise insights about the behavioral relevance of information encoded by neural populations.

## Mapping content-specific encoding and transmission of information within a network

MINT provides both algorithms to study information encoding in individual network nodes and information transmission across nodes. We here illustrate how to combine them for reverse-engineering the information flow within neural networks.

We first simulated a network with four nodes $X_1, ..., X_4$ each modeling the aggregate activity of a brain area (as, e.g., measured by aggregate neural signals such as LFPs, M/EEG or fMRI, see SM 6.3 in S1 Appendix). This network has a well-defined ground-truth flow of information about two independent stimulus features $S_1$, $S_2$ (Fig 4A). Information about $S_1$ is received from the outside by nodes $X_1$ and $X_4$ in a short time window (3–12 ms from simulation start for $X_1$ and 15–24 ms for $X_4$), and is then sent from $X_1$ to $X_2$ and $X_3$ with a 5 ms delay. Information about $S_2$ is received (in the 3–12 ms window) from the outside by $X_2$ which then sends it to $X_1$ with a 5 ms delay. Nodes $X_3$ and $X_4$ exchange information (also with a 5 ms delay) which is not about $S_1$ or $S_2$. To disentangle the information flow, we computed (using the direct method) information encoded or transmitted at each time (in Fig 4 we plot for each node and link the maximal information values over time, but we show in Fig E in S1 Appendix that time-resolved analysis reconstructs correctly the ground-truth information encoding windows and communication delays), and we used MINT's non-parametric permutations tests to identify significant encoding or transmission. Using Mutual Information between individual stimulus features and individual node activity reveals correctly that all nodes have information about $S_1$ and that only $X_1$ and $X_2$ have information about $S_2$ (Fig 4B). To study how this information is exchanged within the network, we first computed overall information transfer with Transfer Entropy, finding correctly significant transfer from $X_1$ to $X_2$ and $X_3$, from $X_2$ to $X_1$, and from $X_3$ to $X_4$ (Fig 4C). To reveal the information content of this exchange we computed Feature Specific Information Transfer (FIT), revealing correctly that the information transferred from $X_1$ is about $S_1$ but not about $S_2$, and that the information transferred from $X_2$ is about $S_2$ (Fig 4D). FIT finds no information transfer from $X_3$ to $X_4$ about $S_1$ or $S_2$, thus determining correctly that the overall information transfer from $X_3$ to $X_4$ detected with TE is not about any of the two stimulus features. Finally, the finding that $X_1$ and $X_4$ encode information about $S_1$ while they do not receive it from other network nodes implies that $X_1$ and $X_4$ receive external $S_1$ information. Similarly, because $X_2$ encodes information about $S_2$ while not receiving within-network $S_2$ information demonstrates that $X_2$ receive external $S_2$ information. Thus, combining encoding with transmission analyses could correctly reverse engineer the within-network specific information flow.

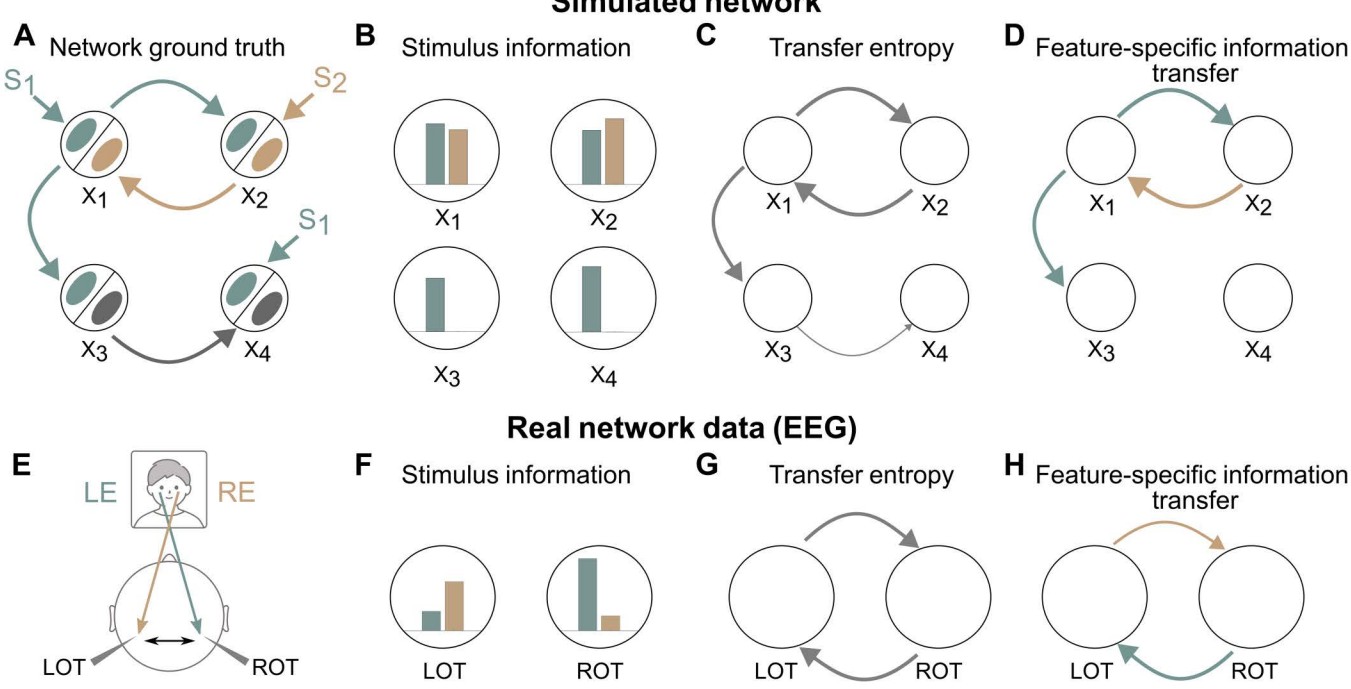

**Fig 4. Reverse engineering information flow using stimulus-encoding and stimulus-transfer estimation algorithms.** Panels A-D test MINT on simulated network data. A: Schematic of the simulation. The network comprises four neural nodes (black circles) $X_1$, ..., $X_4$, each containing two subpopulations (ellipses within the circles) encoding two independent binary stimulus features $X_1$, $S_2$. The ground-truth stimulus specific information communication is plotted in Panel A, with grey color used to indicate no stimulus selectivity, and green and brown colors used to indicate information selectivity to $X_4$ and $S_1$ respectively. B: Maximum Mutual Information across time between each neural population $S_2$ and the stimuli $S_1$ and $S_2$. C: Transfer Entropy (TE) between nodes. D: FIT about $X_i$ and $S_1$ between nodes. In panels C-D, only significant ($p < 0.01$, permutation test) links are plotted, with thickness proportional to the computed value. In each panel we plot the average information values across $n = 10$ simulation repeats. Panels E-F test MINT on real human EEG data. E: Schematic of the putative information flow inter-hemispheric information flow. LOT (ROT) denote Left (Right) Occipito-Temporal regions. LE (respectively RE) denote the Left (respectively Right) Eye face visibility feature. F: Maximum Mutual Information across time about the left or right eye visibility present in left of right OT region. G: Significant Transfer Entropy between LOT and ROT brain regions. H: Significant FIT between LOT and ROT brain regions. In panels G-H, only significant ($p < 0.01$, permutation test) links are plotted, with thickness proportional to the computed value. In each panel we plot the average information values across $n = 15$ experimental subjects. See SM6.3 and SM7.1 in S1 Appendix for details of simulations and real data analysis. Human face sketch is modified from svgrepo.com/svg/493087/men-in-their-20s-and-30s-face, and head sketch is modified from doi.org/10.5281/zenodo.3926093. All resources are under license CC BY 4.0 (creativecommons.org/licenses/by/4.0).

We next tested how MINT reverse-engineers information flow in real brain networks by applying it to an existing EEG dataset recorded from human participants detecting the presence of either a face or a random texture from images covered by random bubble masks [99]. Prior work [36,99,100] revealed that the visibility of the eye region (proportion of visible pixels in the eye area) is critical for successful face discrimination and that the Occipito-Temporal (OT) EEG electrodes are those encoding most Mutual Information about both left and right eye visibility (Fig 4E and 4F). To understand if some of this information was exchanged across the OT regions in different hemispheres, we used TE and FIT to analyze transmission of left or right eye visibility information across OTs. TE across hemispheres was found in both directions (right-to-left and left-right), suggesting a bi-directional inter-hemispheric communication (Fig 4G). However, specific information transfer was precisely directional: FIT about the left eye was only from right-to-left and FIT about the right eye was only from left-to-right (Fig 4H). Thus, using MINT allowed establishing encoding and directional transfer of different eye features across hemispheres with high specificity. These analyses could also temporally localize both encoding and inter-hemispheric transfer (Fig F in S1 Appendix).

Together, these results illustrate the power of combining MINT tools to reverse-engineer encoding and flow of specific information across brain networks.

### Availability and future directions

MINT is downloadable in source code (https://github.com/panzerilab/MINT with DOI doi.org/10.5281/zenodo.13998526), including a Dockerfile, and is licensed under GNU GPLv3. It contains documentation on using it and on building and installing it from source, unit tests, use examples, and replication of paper figures (https://github.com/panzerilab/MINT_figures).

The modularity of MINT allows it to be used alongside any other MATLAB function or toolbox. As exemplified above, we already provide pipelines for interfacing with decoding toolboxes. We plan to add plugins to generate neural and behavioral data from data acquisition and preprocessing toolboxes (e.g., [101]) with MINT's input-data format requirements, and to generate MINT's outputs suitable to be fed directly into toolboxes for further advanced analyses, e.g., for network analysis of information-transfer outputs [102].

We plan to further extend the range of information-theoretic methodology implemented in MINT. MINT's current version emphasizes discretized maximum likelihood estimators. However, we provide only a handful of data-discretization techniques that go with it. We plan to endow them with optimal discretization algorithms based on model selection techniques (Akaike and Bayesian information criterions). While MINT already implements a number of probability estimators for real-valued data we plan to extended them to include other binless and kernel-based estimators [103,104], and parametric probability models (Gaussian, Poisson) proposed in the neuroscience literature. Although we provide several tools for assessing the role of correlated activity, we plan to implement currently missing Maximum Entropy estimators [63]. Finally, the derivation of new neuroscience-related information quantities with PID is highly active [71,105] and the open source and modularity of MINT will allow rapid integration of new developments.

A limitation that may restrict MINT's usage is that it is developed only in MATLAB at this stage. We are thus developing a translated Python version of MINT to widen usage. However, we verified that MINT is usable from Python using the MATLAB Engine API for Python and we provide instructions in SM2 in S1 Appendix.

### Supporting information

**S1 Appendix. Supplementary material.** Theoretical definitions, descriptions of the available methods in the toolbox, detailed descriptions of the simulation specifications used in the figures as well as supplementary analysis and figures. (PDF)

**S1 Data. CA1 recordings in hippocampus.** Calcium imaging recordings used to produce part of the results in Fig 2. The file should be first unzipped with any unzip software and then can be uploaded using the software provided in our MINT toolbox. (GZ)

### Acknowledgements

We thank C. Becchio, G. Iurilli and H. Nili for feedback.

### Author contributions

**Conceptualization:** Gabriel Matías Lorenz, Nicola Marie Engel, Stefano Panzeri.

**Data curation:** Gabriel Matías Lorenz, Nicola Marie Engel, Loren Koçillari, Sebastiano Curreli.

**Formal analysis:** Gabriel Matías Lorenz, Nicola Marie Engel.

**Funding acquisition:** Tommaso Fellin, Stefano Panzeri.

**Investigation:** Gabriel Matías Lorenz, Nicola Marie Engel, Marco Celotto, Loren Koçillari.

**Methodology:** Gabriel Matías Lorenz, Nicola Marie Engel, Marco Celotto, Loren Koçillari, Stefano Panzeri.

**Project administration:** Stefano Panzeri.

**Resources:** Stefano Panzeri.

**Software:** Gabriel Matías Lorenz, Nicola Marie Engel, Marco Celotto.

**Supervision:** Stefano Panzeri.

**Validation:** Gabriel Matías Lorenz, Nicola Marie Engel, Marco Celotto, Loren Koçillari.

**Visualization:** Gabriel Matías Lorenz, Nicola Marie Engel, Loren Koçillari, Sebastiano Curreli.

**Writing – original draft:** Gabriel Matías Lorenz, Nicola Marie Engel, Marco Celotto, Stefano Panzeri.

**Writing – review & editing:** Gabriel Matías Lorenz, Nicola Marie Engel, Marco Celotto, Loren Koçillari, Sebastiano Curreli, Tommaso Fellin, Stefano Panzeri.

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
