## [Decision Letter · Decision Letter 0]

12 Dec 2024

PCOMPBIOL-D-24-01933

MINT: a toolbox for the analysis of multivariate neural information coding and transmission

PLOS Computational Biology

Dear Dr. Panzeri,

Thank you for submitting your manuscript to PLOS Computational Biology. After careful consideration, we feel that it has merit but does not fully meet PLOS Computational Biology's publication criteria as it currently stands. Therefore, we invite you to submit a revised version of the manuscript that addresses the points raised during the review process.

Please submit your revised manuscript within 30 days Feb 11 2025 11:59PM. If you will need more time than this to complete your revisions, please reply to this message or contact the journal office at ploscompbiol@plos.org. Please include the following items when submitting your revised manuscript:

We look forward to receiving your revised manuscript.

Kind regards,

Daniele Marinazzo

Section Editor

PLOS Computational Biology

**Additional Editor Comments:**

As suggested by one of the reviewers, some pointers to other estimators and implementations would be beneficial.

Being this a software paper, a mention of a couple of other toolboxes implementing information theoretical measures, one in matlab https://journals.plos.org/plosone/article?id=10.1371/journal.pone.0109462, and one in python https://joss.theoj.org/papers/10.21105/joss.07360, would help the reader in the navigation of the state of the art.

**Journal Requirements:**

5) Figures: 4E, and S6a, include image of an identifiable person. Please provide written confirmation or release forms, signed by the subject(s) (or their guardian), giving permission to be photographed and to have their images published under a Creative Commons license. You may upload permission forms to your submission file inventory as item type 'Other'. Otherwise, we kindly request that you remove the photograph.

Potential Copyright Issues:

- Figures: 1, 2, and 3; Please confirm whether you drew the images / clip-art within the figure panels by hand. If you did not draw the images, please provide a link to the source of the images or icons and their license / terms of use; or written permission from the copyright holder to publish the images or icons under our CC BY 4.0 license. Alternatively, you may replace the images with open source alternatives. See these open source resources you may use to replace images / clip-art:

7) Please amend your detailed Financial Disclosure statement. This is published with the article. It must therefore be completed in full sentences and contain the exact wording you wish to be published. Please ensure that the funders and grant numbers match between the Financial Disclosure field and the Funding Information tab in your submission form. Note that the funders must be provided in the same order in both places as well. State what role the funders took in the study. If the funders had no role in your study, please state: "The funders had no role in study design, data collection and analysis, decision to publish, or preparation of the manuscript.".

**Reviewers' comments:**

Reviewer's Responses to Questions

**Comments to the Authors:**

Reviewer #1: The MINT toolbox is a great resource for computational neuroscientists. Tools for information-theoretic measures in neuroscience exist in a scattered state and often with little documentation or consistency. This toolbox provides an umbrella for a whole suite of applications with consistent syntax and methods. I found the code easy to install on Windows and very well documented.

I only have some very minor comments, and I leave it at the authors’ discretion whether they wish to include them in a revision.

One note is that I did not see mention of embedding for the transfer entropy. Perhaps a brief mention could help? Occasionally, technical concepts are introduced without much explanation. The toolbox covers a lot of ground so this is to some extent inevitable. It also won’t be an obstacle to many users of the toolbox, who come to it for the software tools rather than to learn about the measures themselves. Indeed, the code documentation itself is very helpful in this regard. However, some additional explanation may prove helpful for novice readers who may not yet have a strong conceptual foundation. For example, the use of SVMs could do with a little more elaboration about rationale and implementation. An example of clarification done right is Figure 4A-D, which I found very clear and extremely helpful for following along. The representation of axis misalignment is also very valuable.

This toolbox will be of great value to the community and I wish to thank the authors for the hard work that clearly went into making and sharing this resource.

Andrea Luppi

Reviewer #2: Review of “MINT: a toolbox for the analysis of multivariate neural information coding and Transmission” by GM Lorenz et al.

This paper describes a toolbox which is in my view an extremely important to quantitative neuroscience. There has been continuing interest, over the years, in information analysis of neurophysiological datasets, with consequent demands for toolboxes to compute these quantities. More recently, there has been substantial interest in computing some slightly more complicated information theoretic quantities, due in particular to the popularisation of partial information decomposition. The challenge is that these quantities can be more difficult to estimate, with at least one widely used public estimation toolbox falling back on “plugin” estimation of entropy – meaning that scientific uses of these tools are likely subject to major pitfalls. The MINT toolbox, which implements a number of advanced information theoretic quantities while at the same time using well-tested and efficient estimators of entropy as their basis, is thus extremely welcome.

The package is demonstrated with application to a diverse set of interesting real world and simulated examples. Some of these examples, however, might better other estimation approaches than those adopted here (see specific points below) and thus I recommend a few small additions to the toolbox in order to make it of greater applicability to a wider range of potential neuroscience users.

Specific points

• The relatively limited number of estimators provided may limit the applicability of the toolbox. For instance, the Kraskov estimator has been of widespread use in the community for information calculation, is one of the best performing estimators for continuous data (see Holmes & Nemenman PRE 20189 for a good discussion), and in particular is used by many for PID calculations. The choice of which estimator to use will depend upon the nature of the dataset analysed – for instance continuous vs discrete/spiking, and in the latter case depending on sparsity of activity. While there are several estimators provided, they are mostly of rather niche utility (eg Ish) and make a rather electic set. I suggest that the authors strongly consider implementing the Kraskov (KSG) estimator (ideally the improved version due to Holmes & Nemenman) alongside the direct method estimation currently present, in order to broaden the utility of the package from niche to general use. And certainly the paper needs a (much) more detailed discussion of entropy and information quantity estimation in the main text, including the provision of advice on when different estimation approaches may be useful.

• The tools for dimensionality reduction using decoders is a feature which is absent in most information toolboxes, so this is good to see. It is not clear to me how this interplays with choice of information estimation strategy, and this should be discussed. For instance, I would a priori expect that the discrete estimation strategies adopted here might be suboptimal in these circumstances, and the improved KSG estimator (Holmes & Nemenman) might be best in these circumstances? I am happy to be proved wrong, but I think the paper should provide some validation that the approach is the best one here. The paper demonstrates that it works on the example provided – but you can always find a harder / higher dimensional example, and it would be good to see that it works better than other approaches.

• To extract stimulus-related information, the toolbox uses differences between interaction information in two conditions (stimulus present vs absent or correlated vs independent). However, II is a non-directed information measure that captures the overall synergy between the groups of variables. Therefore, it necessarily requires the user to take a difference with a control condition to isolate the synergy/redundancy the user cares about. In the absence of such controls, the toolbox provides PID measures to extract directed synergy/redundancy. However, a directed version of II known as the Redundancy-Synergy Index (RSI) can provide the benefit of not assuming a redundancy function while characterizing the balance between synergy and redundancy of the system. We suggest implementing RSI as a logical suggestion for completeness, but not a necessity.

• The toolbox aims to be universally applicable to different modalities of data (EEG,fMRI,Spiking etc), by relying on discretisation of the signal. Then, using discrete information estimators. This method although useful in information-theoretic estimation of continuous signals has many limitations. Especially when the temporal structure of continuous signals may be implicated. Different binning protocols may be useful for different kinds of analysis. Such limitations should be discussed especially in case of continuous signals such as M/EEG and fMRI.

• It is in some ways unfortunate that the toolbox is implemented in MATLAB, as this language is now falling out of use among the computer-literate neuroscience communities (and even engineering communities) that form the target audience. A python implementation would have been likely to find much more widespread use. The authors should comment on how the system could be used from python using for instance the MATLAB Engine API for Python.

• Overall, limitations of the toolbox, i.e. under what conditions it is not applicable, should be discussed. I would suggest a longer discussion section in which advice is provided about what the toolbox can do well in its current form, what it can do but may not be ideally configured for, and what still needs to be implemented .

• Figure 1, panel E, “plague” is exactly how it feels, but I suggest the choice of a less loaded term.

Very Minor

• Some typos, eg “empirically occurrences” (p. 4). The manuscript would benefit from a pass to correct these.

Reviewer #3: In this manuscript, the authors describe a set of computational methods, implemented in MATLAB, for computing a number of information theoretic measures that are relevant to interpreting behavioral neuroscience data. These include measures that can be used on single cells and pairs of cells (including those that look for synergistic encoding and those that measure transfer of information across time), as well as methods for incorporating larger groups of neurons by implementing basic dimensionality reduction, both unsupervised (e.g. PCA) and supervised (e.g. SVM). The authors usefully present a few examples of applying the methods to real and simulated datasets.

The manuscript is for the most part clearly presented, and the existence of such a toolbox will no doubt be beneficial to the field. The lack of a clear consensus in the literature of the most useful measures means that there are other related measures that are not represented here (e.g. other ways of dealing with small-N bias such as the NSB method; other ways of computing synergy among larger sets of neurons such as the O-information)--- but the included set of measures is large enough that this is easily more complete than any other existing software package.

Some relatively minor concerns:

— I was confused by the Fig 3C left plot. Is this just an example of one particular pair of neurons?

— In the simulated example in Fig. 4, what is the particular dynamic of “sending” information? E.g. what is the timescale of information transfer, and what needs to be known about these timescales for this to work in analyzing real data?

— A few typos: "will lead to uncover numerous new insights"; "as empirically occurrences"; "directly form"

**Have the authors made all data and (if applicable) computational code underlying the findings in their manuscript fully available?**

Reviewer #1: Yes

Reviewer #2: Yes

Reviewer #3: Yes

PLOS authors have the option to publish the peer review history of their article (what does this mean? ). If published, this will include your full peer review and any attached files.

**Do you want your identity to be public for this peer review?** For information about this choice, including consent withdrawal, please see our Privacy Policy .

Reviewer #1: No

Reviewer #2: **Yes: ** Simon Schultz

Reviewer #3: No

**Figure resubmission:**
---

## [Decision Letter · Decision Letter 1]

6 Mar 2025

Dear Professor Panzeri,

We are pleased to inform you that your manuscript 'MINT: a toolbox for the analysis of multivariate neural information coding and transmission' has been provisionally accepted for publication in PLOS Computational Biology.

Best regards,

Hugues Berry

Section Editor

PLOS Computational Biology

Hugues Berry

Section Editor

PLOS Computational Biology

Reviewer's Responses to Questions

**Comments to the Authors:**

Reviewer #1: I wish to thank the authors for constructively addressing all my comments, and for a very valuable addition to the literature.

Reviewer #3: The authors have successfully responded to all of my comments.

My only suggestion relates to the new Figure C in the supplemental material: I like this figure a lot in that it (1) shows explicitly how dimensionality reduction is useful and (2) gives a reader a rough order of magnitude intuition for how much data is needed to get reasonable estimates. I know the manuscript is already pushing up to length limits, but the authors might consider squeezing some of this into the main text.

**Have the authors made all data and (if applicable) computational code underlying the findings in their manuscript fully available?**

Reviewer #1: Yes

Reviewer #3: Yes

PLOS authors have the option to publish the peer review history of their article (what does this mean? ). If published, this will include your full peer review and any attached files.

**Do you want your identity to be public for this peer review?** For information about this choice, including consent withdrawal, please see our Privacy Policy .

Reviewer #1: No

Reviewer #3: No

---

## [Editor Report · Acceptance letter]

PCOMPBIOL-D-24-01933R1

MINT: a toolbox for the analysis of multivariate neural information coding and transmission

Dear Dr Panzeri,

I am pleased to inform you that your manuscript has been formally accepted for publication in PLOS Computational Biology. Your manuscript is now with our production department and you will be notified of the publication date in due course.

With kind regards,

Lilla Horvath
